# Conceptualizing safer sex in a new era: Risk perception and decision-making process among highly sexually active men who have sex with men

**Nicklas Dennermalm**[1,2]\*, **Kristina Ingemarsdotter Persson**[1], **Sarah Thomsen**[1], **Birger C. Forsberg**[1], **Helle Mølsted Alvesson**[1]

**1** Department of Global Public Health, Karolinska Institutet, Stockholm, Sweden, **2** Department of Social Work, Stockholm University, Stockholm, Sweden

\* nicklas.dennermalm@socarb.su.se

## Abstract

### Background

Men who have sex with men (MSM) are at the epicenter of the HIV epidemic. Efforts to prevent sexually transmitted infections (STIs) and HIV transmission have traditionally focused on condoms and abstinence from high risk sexual practices. Recently, additional methods such as pre-exposure prophylaxis (PrEP) and viral load sorting have been introduced. The aim of this study was to gain understanding about risk management and risk perception strategies for HIV among highly sexually active Swedish MSM with men in Berlin.

### Methods

Eighteen sexually active Swedish MSM who travelled to or lived in Berlin were recruited and interviewed in this study. The data were analyzed using content analysis.

### Results and discussion

These men represent a group of knowledgeable MSM in terms of HIV. They acknowledged that having sex with men in Berlin was linked to high sexual risk taking due to the higher prevalence of HIV/STIs than in Sweden, but reported that they nevertheless did not alter their risk management strategies. The analysis resulted in a conceptual model of risk assessment that allows for a deeper understanding of the complexity of the risk reduction decision-making process. Three ontological perceptions of risk were identified: accepting, minimizing and rejecting risk. Seven practiced risk reduction methods were described. Some informants applied their preferred method or set of methods to all settings and partners, while others faced complex decision-making processes.

### Conclusion

HIV is integrated into the core of MSM's sexuality, independently of how they ontologically related to the idea of risk. A constant navigation between pleasure, risk and safety,

**Data Availability Statement:** All data supporting our work is provided within the manuscript.

**Funding:** The project was funded by The Public Health Agency of Sweden, reference nr 01011-2017-2.3.2, with BF as principal recipient and ND and HMA as co-recipients. Co-authors KIP and ST received no funding for their contribution of this article. The funders had no role in study design, data collection and analysis, decision to publish, or preparation of the manuscript.

**Competing interests:** The authors have declared that no competing interests exist.

alongside having to relate to risk created a complex process. Efforts were made to remove HIV from their lives by rejecting the idea of risk, and thereby reject the idea of the homosexual body being a possible vessel for a virus and an epidemic.

## Introduction

Gay and bisexual men, and other men who have sex with men (MSM) have been disproportionately affected by the HIV epidemic since the 1980s [1–3]. MSM have a higher prevalence compared to the general population also in high-prevalence countries in the global south [4]. Thus, this group has been targeted with a variety of public health interventions ranging from the closing of gay saunas [5], safer sex leaflets, some emphasizing fear as a tool [6–8] and free condoms [9], to abstinence marketing [10], pre-exposure prophylaxis for HIV (PrEP) [11–15] and, mostly in African contexts, male circumcision [16, 17]. The epidemic has influenced and is still affecting the MSM community through stigma and new infections. It has also left its mark on art and popular culture, including works by playwright Tony Kushner [18] and the television series Pose. The character and perception of the epidemic has changed over time, from being chaotic, stigmatized and deadly, to a chronic but treatable infection [19]. Furthermore, research on undetectable viral loads [20–22], together with the introduction of PrEP [23, 24] or informal use of PrEP [25] are measures that have changed the HIV prevention landscape. PrEP can be seen not only as a tool to reduce risk for HIV transmission, but also to develop agency, increase empowerment, opposing normative discourses surrounding HIV and sexuality among MSM and increase pleasure, intimacy and quality of life [13]. However, the stigma remains, even without the visible signs and consequences of AIDS. Herek explains this by separating 'instrumental AIDS stigma' defined as the risk and fear or communicability and lethality of HIV from the 'symbolic AIDS stigma,' meaning the symbolic meaning of AIDS and its proximity to homosexuality and substance use [26].

Risk is often seen as something to avoid due to its per-definition negative outcome [27, 28]. Beck (1992) described how the public and politics entered the private sphere, such that risk was no longer only in relation to one's own health, but in a "risk society" it also had social, economic and political consequences [29]. The social construct of risk has been addressed in numerous works. Douglas and Calvez argued that society´s majority view on risk neglects the fact that society consists of many different cultures and sub-cultures that do not all have the same perception of risk or assessment of different kinds of risk [30]. Later research builds upon this and argues that "to call something a 'risk' is to recognize its importance to our subjectivity and well-being" and that such recognition differs between contexts and cultures [31]. Lupton and Tulloch state that the overall social construct of risk is defined by the "dominant culture," but may be perceived in a different way in sub-cultures or smaller groups of people [32].

Lupton and Tulloch introduced what they call "the discourse of emotional engagement" [32]; that risk is something which may be associated with positive connotations that elevate an experience, be it having sex or something else. Such emotions can be generated by feelings of being closer to danger, belonging to a community, or breaking the rules. A person may explore an overall elevated bodily experience of being "swept away" and "closer to nature than culture" in addition to disobeying rules [27]. These researchers also relate risk and risk management to a Western discourse on "the ideal of the civilized body" that aims at control and regulation of both the self and bodily pleasures [32]. This is closely linked to Michel Foucault's work about

the discipline of the body, paraphrasing Plato by stating that the soul is the prison of the body [33]. The civilized body listens and acts on reason and changes its behavior in order to stay healthy. However, "where there is power, there is resistance" [34] and previous researchers have noticed a significant resistance to health promotion among gay men [35]. Lupton and Tulloch described a will to deliberately escape the ideal of the civilized body, to escape from cultural control and reject behaviors benefitting health that are desired by society. Instead, there is the will to focus on bodily pleasure, which may stem from the presence of risk, the forbidden and the "contaminated" [32]. Lupton discusses the common perception that individuals taking risks or rejecting risks do so due to ignorance or irrationality [31]. Lindroth suggests that what is interpreted as risk can better be understood as chance, and by using the terms risk-taking and chance-taking instead of risk behavior or risk-taking behavior, one can address a person's rational actions; in sexual situations, chance-taking can outdo risk by offering rewards such as pleasure, intimacy, a relationship or even love [36]. This understanding of rationality and risk may be applicable to the understanding of sexual risk taking in MSM.

A broad range of issues are relevant in order to understand sexual risk taking behavior, including psychological, social, legal, emotional, cultural, intellectual, situational and moral issues [37]. Sexual decision making is complex and previous research has suggested that it should be subjected to further attention from both researchers and practitioners [13, 38–40]. For heterosexual relations, sexual decision making has historically included concerns such as gender norms, STIs and pregnancy risk. For MSM, HIV transmission risk is high due to practices of anal sex without a condom [41]. Sexual decision making among MSM has been described as a dynamic process, in which aspects like PrEP and context and assessment of the partner's risk taking with others are part of a well-defined larger strategy of staying HIV negative. The tool used to stay negative is usually not an isolated, single solution [39]. Understanding and classifying risk includes multiple factors affecting the actual risk, rather than being based on single parameters [42]. MSM have been characterized as being divided into three different risk trajectories, low, medium and high-risk groups, depending on how decisions affect their behavior [42]. Furthermore, people may change their level of risk over time due to changes in drug use, state of well-being and other factors [39, 42]. The distinction between the epidemiology of HIV risk and the psychological experience of decision making with regards to risk has also been highlighted [43] and some research suggest that rational decision based on probability estimates are subordinate pleasure-seeking [13]. Most MSM who have condomless anal intercourse (CLAI) fit into three main categories: 'intimates' (CLAI only with main HIV negative partner), 'trusters' (CLAI only with casual partners believed to be HIV negative) and 'gamblers' (CLAI with unknown HIV status). In one study the experience of anal intercourse without a condom was found to be the same in these groups, but the perception as well as the actual risk of being low or high was different [43]. The decision-making process has also been described as semiotic and built into the search for intimacy, pleasure and existential perspectives, as well as public health and corporate neoliberal rhetoric and autobiographical narratives [44]. Pleasure is a complex component which should be considered and understood within HIV prevention [13]. Previous research include the link between seamen, pleasure and intimacy [45], barriers for increased pleasure and PrEP use due to the stigma of being a 'Truvada whore' [46] and pleasure and intimacy among black MSM [47] and pleasure-driven impulsivity [48]. Research has also explored the hypothesis of condom fatigue and proneness to stop using condoms once HIV became treatable, also known as 'AIDS optimism'or 'HIV optimism,' with various outcomes depending on identified sub-groups of MSM [44, 49–52].

Sweden is a low-prevalence country with high access to treatment and claims to be the first country to reach UNAIDS' goal of 90-90-90 [53]. The latest prevalence estimate showed a 7% prevalence among MSM [54] with approximately 8020 (2019) people living with HIV [55].

The quantitative Swedish research on MSM and HIV is rigorous [54, 56–62]. In contrast, qualitative Swedish research on MSM and HIV is scarce [63], highlighting the need for increased effort within the research community to both address Swedish perspective but also to recognize Swedish MSM's part in a European setting.

The aim of this study was to gain understanding about risk management and risk perception strategies among highly sexually active Swedish MSM in order to understand their decision-making process.

## Method

### Study design

This research is part of a larger qualitative project exploring travel, drug use and sexual health among highly sexually active Swedish MSM who spend time in both Sweden and Berlin [64, 65].

### Recruitment

The participants were recruited using network sampling [66]. The eligibility criteria were: i) Swedish citizen, ii) cis-gender men who have sex with men, iii) aged 18–46 and iv) currently or formerly a resident of Berlin or travels to Berlin at least twice per year. In this study each 'seed' contributed with one, or by preference, two referrals to minimize the risk of bias due to the initial seed being more likely to contact people both would know. Three initial seeds were recruited from the research team's network and two were added later in the process. The informants were compensated with two movie tickets for their time spent in the interview. No reward was given for providing referrals.

### Participant characteristics

The sample consisted of 18 Swedish MSM, 23–44 years old. All were current or former residents of Berlin or travelled to Berlin at least twice a year. They had visited both Berlin and Sweden and sought partners in both places over the past three years. At the time of the interview, seven identified as singles, eight were in open relationships, two were married but the relationship was sexually open, and one was unsure of his relationship status. The majority had attended university and were also employed. Sixteen had had one or more STIs and one had contracted HIV. Overall, the men had high numbers of partners in Sweden and the number increased further in Berlin. The men interviewed also described experiencing a broad range of sexual practices in both cities, such as sex on the premises of club venues, group sex, and fisting. Fifteen of the men experienced using drugs in club and/or sex settings [65]. The names used in the Results section are fictitious.

### Data collection

Semi-structured open-ended interviewing was used as the data collection method between January 2016 and June 2017. The interview guide was designed based on themes generated from an analysis of the Swedish MSM2013 survey [57], as well as specific themes to gain deeper understanding of the person's life at the time of the interview. Questions were asked about biographical data, reasons for travelling to Berlin, purpose of dating, dating platforms, sexual culture, HIV/STI, safer sex, HIV/STI testing, alcohol, drugs, and living with HIV. The interview guide was piloted with three participants with only minor alterations, thus the pilot interviews were included in the study. Interviews were conducted in private either face to face or over video, in the interviewer's office, interviewer's home or at the home of the interviewee. The

interviews lasted 45–170 minutes. The informants were not offered the opportunity to read their transcripts or provide feedback on the findings. However, those requesting to read the transcripts were granted that access. The research findings were distributed to those who requested it. The first author kept an anonymized research diary during the research process, which was shared within the research group and used for initial discussions of the data.

## Analysis

Qualitative content analysis was used [67]. The coding and clustering into sub-categories, categories and themes was executed as an iterative process in order to gain deeper understanding of the data. NVivo and Excel were used for coding and creating structure. A second researcher listened to and read the transcripts, as well as reviewed the coding process and provided an alternative understanding and interpretation of the data. The initial coding process and the final clustering was done under the supervision of a senior qualitative researcher. The concept of theoretical saturation was discussed and was judged to have been reached for all interview guide topics after the completion of 18 interviews, as only small variations emerged in the new codes that no longer altered the themes and patterns.

## Ethics

The study's ethical approval was granted by Ethical Review Board in Stockholm, reference number 2016/32-31. The current review authority is the Swedish Ethical Review Authority. The research project complied with the German Data Protection Act (BDSG, 20.12.1990) and the Berlin Data Protection Act (BlnDSG, 3.7.1995) to protect the integrity and safety of the participants. The men in the study were informed both verbally and in writing beforehand about the project, the aim of the study and the procedure of confidentiality. A letter of consent was sent to the men the day before the interview and discussed and signed before the interview started. The men could at any time withdraw from participation, which no one did. The audio files and anonymous transcriptions were stored on locked devices. The audio files were only shared between two of the researchers. Potential distress was addressed verbally and in writing, with referral to established professionals free of charge. Aligned with the sampling method, the men provided the researchers with the contact information of new informants, but only if they accepted the information to be shared. No information regarding the participation of the new informants were given to the original informant. We have no information on how many chose not to accept the invitation.

## Reflexivity

The researcher's position may influence different aspects of the research process. Having an interviewer who is perceived to be sympathetic to the context may have an increased access to the cultural contexts and making the participants more willing to be interviewed [68]. Also, the interviewer's position may shape the willingness to share [69]. The interviewer was, beside a Masters student of global public health, a Swedish gay man with an extensive professional background in MSM's sexual health at a well-known non-governmental organisation, which we believed contributed to building trust as well as increasing the willingness of the participants to share information as well as recruit new informants. With that in mind, investigating MSM forced the research team to reflect upon the dynamics of emic and etic perspectives and how the interviewer could benefit from both perspectives. The balance was upheld using two main measures: review of the transcripts and coding a few weeks after the interview with a 'new lens' and peer consultation from a senior co-author in regards to data collection, coding and analysis (ST) [69].

## Results

Content analysis resulted in several themes. The theme explored in this article is *risk perception and risk management* with regards to HIV and STIs. In the reporting of the results the focus is on a description of manifestations of this theme rather than interpretation of latent components.

### Risk perception

The majority of the men perceived Berlin to be a city with a higher prevalence of HIV and STIs than Sweden. However, one of the men stated that he had no idea about the epidemiological situation in Berlin and added that statistics are not helpful when protecting oneself against HIV.

The attitude towards contracting an STI ranged from not caring, to concern or worry. Concern were expressed over not being able to cure gonorrhea in the future due to drug resistance; a fear at both the individual and community level. The majority of the informants perceived themselves to be HIV negative. With regards to how it would feel to be HIV positive, attitudes spanned from a wish to not get it, to angst. None of the men expressed indifference on this matter, even though some men expressed no fear about becoming HIV positive. "I'm done with the drama. Getting HIV wouldn't be a big thing if I got it now. Still, I don't want it though. . ." ('Nils'). The men mentioned concerns about side effects, the chronic condition and other reasons for not wanting to become HIV positive. Some linked HIV to stigma in various ways. "People are still unaware of [HIV]. I believe many people are afraid to date someone with HIV. That is what I am afraid of the most, to have HIV and be alone for the rest of my life" ('Nofri'). One respondent criticised the built-in options to show one's HIV status in dating apps "as it becomes some kind of stigma, if I cannot present a certificate of being [HIV negative], then I am not credible" ('Dennis').

The main purpose of condoms and other risk management methods was to minimize the risk of HIV/STI transmission. The men knew that condoms fulfilled the purpose of minimizing the risk of HIV transmission. However, for some informants, other ideals were more important. The accepting or rejecting of risk was not only theoretical but real, not necessarily every time they had sex but frequently at different time periods.

The idea of risk in association with sex played a large role in the sex lives of some informants. It was not risk in terms of something exciting or desirable, but rather something that had become an integral and dominant part of sexuality.

> When you think about sex, when you want to have sex or when you are speaking to friends about it, there is always the infection perspective in the back of your head. (. . .) I think about [infections] before [sex], maybe during it and afterwards I think about it. ('Ulrik')

Sickness and infections were dominant parts of the majority of the sex lives of these informants; when looking for sex, choosing partners, deciding upon sexual practice and feelings afterwards. The presence of risk and the use of condoms was perceived as more or less unproblematic by some men. By others, it was something highly problematic. Public health messages directed at MSM were seen as patronizing by some men. "But if you have to [affirm your sexuality], make sure that you are goddamn covered in rubber" ('Mårten').

This constant presence of risk did, for some men, create a counter reaction in which they ignored the risk to a certain extent, for some in theory while for others in practice. Several of the men expressed a will to prioritize pleasure instead of risk.

I wasn't willing to stop anything that was about to happen that I wanted to happen, and I said to myself: 'This will be pleasurable.'. (. . .) I wanted to live without thinking of risk for a while ('Izaan').

There were informants who described the concept of pleasure and risk in terms that were more on an ontological level.

Having sex is always taking a risk, risk is part of it. Part of living is the risk of dying. (. . .) 'This is my level of risk', then I decide that this is my starting point and when it's decided upon, you have to be prepared that you are subjected [to risk]. I'm using a condom but it may break ('Nils').

This informant used a condom with no exception but still accepted that the risk of getting HIV was always present. At the same time, he had to come to terms with that level of risk. Another man used condoms less often and sometimes rejected the idea of risk despite the fact that he perceived CLAI as dangerous with regards to HIV/STIs.

. . . sex shouldn't be about disease and potential death. There is a desire in me to stop giving a damn and do whatever I feel like, to embrace lust instead of subjecting to risk. . . ('Mårten').

## Risk management

The vast majority of the participants broadened their range of sexual practices in Berlin compared to when being in Sweden, but their risk reduction strategies were described as the same in both settings. For some informants, risk taking in Berlin generated more concerns afterwards than risk taking in Stockholm, however, the increased concern did not alter behavior. The participants described a variety of risk reduction *methods* used for avoiding contracting HIV or STIs. They were used to different degrees, alone or in combinations to form their overall strategy. However, words like 'method' or 'strategy' were rarely used by the men interviewed but are part of our conceptualization.

The seven risk reduction methods mentioned by the men were: (1) Condom use, (2) Avoiding ejaculation inside the body, (3) Avoidance of situations where your perception was compromised, (4) Partner sorting, (5) Sero-sorting, (6) Viral load sorting and (7) PrEP. See Table 1.

(1) Participants reported having used condoms always or sometimes during anal sex, since they felt it protected them from HIV and/or STIs. Several of the men were satisfied with their current risk management strategy in which the condom was the sole method used or in combination with other methods. The men who used condoms in Sweden also did so in Berlin. Those who had anal sex without a condom did so in both Sweden and Berlin. CLAI was seen by some men as more pleasurable, more intimate and more natural than sex with a condom. For one informant the decision whether to use a condom or not was often made in the moment of action with different degrees of assumptions and preconceptions, and sometimes under the influence of drugs:

They are usually pretty fast decisions. They are not very well thought through, I would say. Unless it's something that has been discussed beforehand like 'Do you know your status?' (. . .) It could have been a discussion like that before the decision was made to have

**Table 1. List of the seven risk management methods described by each study participant.**

| Nr | METHOD 1: Condom during anal sex with non-monogamous partner Yes/No/Inconsistent | METHOD 2: Avoiding ejaculate inside the body Yes/No/Inconsistent | Method 3: Avoidance Yes/No/Inconsistent | METHOD 4: Partner sorting Yes/No/Inconsistent | METOD 5: Sero-sorting Yes/No/Inconsistent | METHOD 6: Viral load sorting Yes/No/Inconsistent | METHOD 7: On PrEP? Yes/No |
|---|---|---|---|---|---|---|---|
| 1 | Inconsistent | No | No | No | No | No | No |
| 2 | Yes | No | Yes | Yes | No | No | No |
| 3 | Yes | No | No | No | No | No | No |
| 4 | Yes | No | No | No | No | No | No |
| 5 | Inconsistent | No | No | No | No | No | No |
| 6 | Inconsistent | No | No | No | No | Inconsistent | No |
| 7 | Yes | No | No | Yes | No | No | No |
| 8 | Inconsistent | No | No | No | Inconsistent | Inconsistent | No |
| 9 | Inconsistent | No | No | No | Inconsistent | Inconsistent | No |
| 10 | Inconsistent | Yes | No | No | No | Yes | No |
| 11 | Inconsistent | No | No | No | No | Yes | No |
| 12 | Yes | No | No | Yes | No | No | No |
| 13 | Inconsistent | No | No | No | No | No (living with HIV) | No |
| 14 | Inconsistent | No | No | Yes | No | No | No |
| 15 | Yes | No | No | Yes | | No | No |
| 16 | Inconsistent | No | No | Yes | Inkonsistant | No | No |
| 17 | Inconsistent | No | No | No | No | No | No |
| 18 | Inconsistent | No | No | No | No | Yes | Yes |

unprotected sex. But usually it is based on weaker assumptions than that. You simply make the decision in the moment: 'What the fuck. It's probably fine' ('Mårten').

The respondent's non-condom-use was not followed by regret or post-hoc rationalizations, despite a perception that condomless anal sex was "dangerous". The decision was made in the heat of the moment; the concept of adapting a more flexible overall strategy was well thought through. He knew and accepted that every sexual encounter could result in either the use of condoms or not. The one informant who described a lower use of condoms in Berlin used PrEP there, which he did not use in Sweden to the same extent.

(2) One method of decreasing the risk of STI acquisition when having receptive CLAI was that the partner had to avoid ejaculation inside the body. One man who preferred receptive anal sex without a condom usually asked his partners not to ejaculate inside him early on during sex, but there were different levels of difficulty depending on if he had met the sex partner (s) before or if he had sex anonymously with partner(s) in a darkroom:

I tend to repeat it a few times while we are at it. Sometimes people can get quite eager, so you never know. (. . .) But it's people you know and you respect each other. But I think it would be more difficult to ask if you are in a darkroom at [name of a fetish club] That is why I try to use condoms in darkrooms ('Mathías').

(3) Avoiding legal and illegal drugs or being in control of excessive use, which could compromise perception, were used by some informants as risk management strategies. One man avoided looking for sex or having sex in sexual settings such as darkrooms where he could not

see the partner clearly, as a visual examination of a potential partner was perceived helpful in risk assessment.

(4) Another method used was partner sorting based on a variety of parameters. Sometimes the sorting was based on stated facts, such as a man stating he was on PrEP in his on-line profile; other times the sorting was more speculative and conclusions were drawn based upon assumptions. The informants described and perceived groups such as people on PrEP, drug users,"barebackers" and "people with an unhealthy sexuality" ('Tobias') as associated with higher risk taking. Therefore, these persons were avoided as sex partners.

Several informants also mentioned people living with HIV. Some claimed that they avoided having sex partners who had HIV because of the perception of persons with HIV being interpreted as a sign of low condom use and therefore higher risk of having STIs. Some informants described avoiding sex partners living with HIV despite knowledge about undetectable viral load. Other informants stated the opposite, that people living with HIV were associated with *lower* risk of STI transmission due to their mandatory regular health check-ups. Similar perceptions existed regarding regular PrEP users. Several of the informants knew that sorting based on perception was not an effective tool in order to avoid HIV and/or STIs, but used it anyway in order to provide emotional comfort.

(5) Sero-sorting usually denotes anal intercourse without a condom with partners of the same HIV status as yourself. For some of the HIV negative study participants this was a strategy to stay HIV negative. The sero-sorting was based on assumptions and/or asking about the partner's HIV status. Others criticized this method: "One shouldn't think, well negative means that everything is fine. Actually, negative means warning" ('Dennis').

(6) Viral load sorting, meaning CLAI with sex partners living with HIV with an undetectable viral load, was mentioned as an opportunity to have CLAI with no concerns of HIV transmission. The concept of undetectable viral load equaling untransmittable (U = U) was known by the vast majority of the men. For several men it was also practiced for risk reduction. This method was fairly new, dating one to three years prior to the time of the interviews. The change of behavior was motivated by new scientific evidence, which they had read about and talked about with friends and/or partners.

> If you are HIV negative and meet [a HIV positive partner with undetectable viral load] you get happy, since you know you can have lots of unprotected sex with that person. It is so much more pleasurable to have unprotected sex ('Johannes').

Some of the informants raised concerns regarding if an undetectable viral load really meant a decrease in risk of HIV transmission, and if changes in viral load could affect the risk of HIV transmission. One concern about having CLAI with people with undetectable viral loads was that the lack of condom use would increase the risk of getting an STI, rather than an increased risk of HIV.

(7) The informants were aware of the concept of PrEP. However, no PrEP programs were implemented at the time of the data collection. Only one man with inconsistent condom use experienced using PrEP. He got the pills from HIV negative friends who ordered them online or from HIV positive friends who had it as part of their medication.

> [I] think it is a very good protection and I wish it was available. (. . .) You'll never get everyone to use a condom every single time anyway. But if you take PrEP, you can at least get rid of the HIV stigma. Sure, you can't cure it and there's still gonorrhea present (. . .) but it feels like HIV is what you are afraid of. Just because it's incurable. With PrEP, I can continue having sex and remove the worst [fear] ('Lars').

This informant used event-based dosing since his supply was limited; he used it when going to Berlin in order to stay safe. He admitted that using PrEP in Berlin implied using less condoms there compared to when having sex in Sweden. He did not get follow-ups on his liver or kidney function but he tested for HIV/STIs regularly. He stated that he would do check-ups if he was on daily PrEP.

PrEP was also described as an extra layer of protection to prevent HIV transmission when condoms, for different reasons, were not used. One man expressed indecisiveness and talked about worries that PrEP would have a negative impact on his sexuality with sex being associated with emptiness. Still, he was motivated to use PrEP as an act of solidarity for the community:

> I don't quite have that [bareback] preference. (. . .) But on the other hand, if it would decrease that whole HIV stigma. (. . .) And if we could help a lot of people not getting HIV, that I would not get HIV. Then what the fuck, give it to me ('Alex').

There were several concerns among the participants about taking PrEP themselves; these included fear of developing resistance towards the drug, the effectiveness of the drug, ability to maintain high adherence, side effects and increased STIs such as multi-resistant gonorrhea.

Despite a general positive perception of PrEP at a community level, several of these highly sexually active men were not interested in taking PrEP themselves since they were satisfied with using condoms as their main risk reduction strategy. The ones with regular condom use tended to be less interested in PrEP than ones with inconsistent condom usage. The men accepting and rejecting risk were more interested in PrEP and viral-load sorting than men striving to minimize risk. These methods were believed to not compromise pleasure and other desired positive effects of condomless sex, while also having protection against HIV transmission.

## Discussion

The men who participated in this study were gay men whose sexual perception and practices were closely linked to the concept of safer sex in the aftermath of the AIDS crisis of the 1980s and 1990s. HIV and the stigma that surrounds the issue still motivated the men to act to minimize the risk of HIV transmission or to keep the risk low. It could be interpreted as an aftershock of the lethality of the 1980's HIV crisis, what Herek meant when he described 'the instrumental fear' of HIV [70]. Despite HIV being less of a crisis compared to previous decades, we found that it still shapes the core of MSM's sexual practices. The metaphoric risk management equation that many of the men tried to solve was that a higher degree of safety meant a lower degree of pleasure. The physical health risks could be high but the chance of sexual pleasure and new experiences trumped this [36, 44, 71]. Previous research addresses this as 'AIDS optimism' and 'HIV prevention fatigue' [44, 49–52]. Our participants felt the urgency to address HIV, stigma and gonorrhea and reported they were not tired of HIV prevention. What they were tired of was having to deal with infections as integral parts of their sex lives. Research has evolved to show that an undetectable viral load means zero risk of HIV transmission [22, 72]. This gives more reasons for HIV negative MSM not to fear HIV, as well as viewing people living with HIV with undetectable viral loads as safe partners from an HIV perspective.

The men's decision-making processes varied greatly and were influenced by perceptions of pleasure and risk [13]. The processes also differentiated in complexity, number of methods used and attitudes towards when to apply which method. We have attempted to visualize this

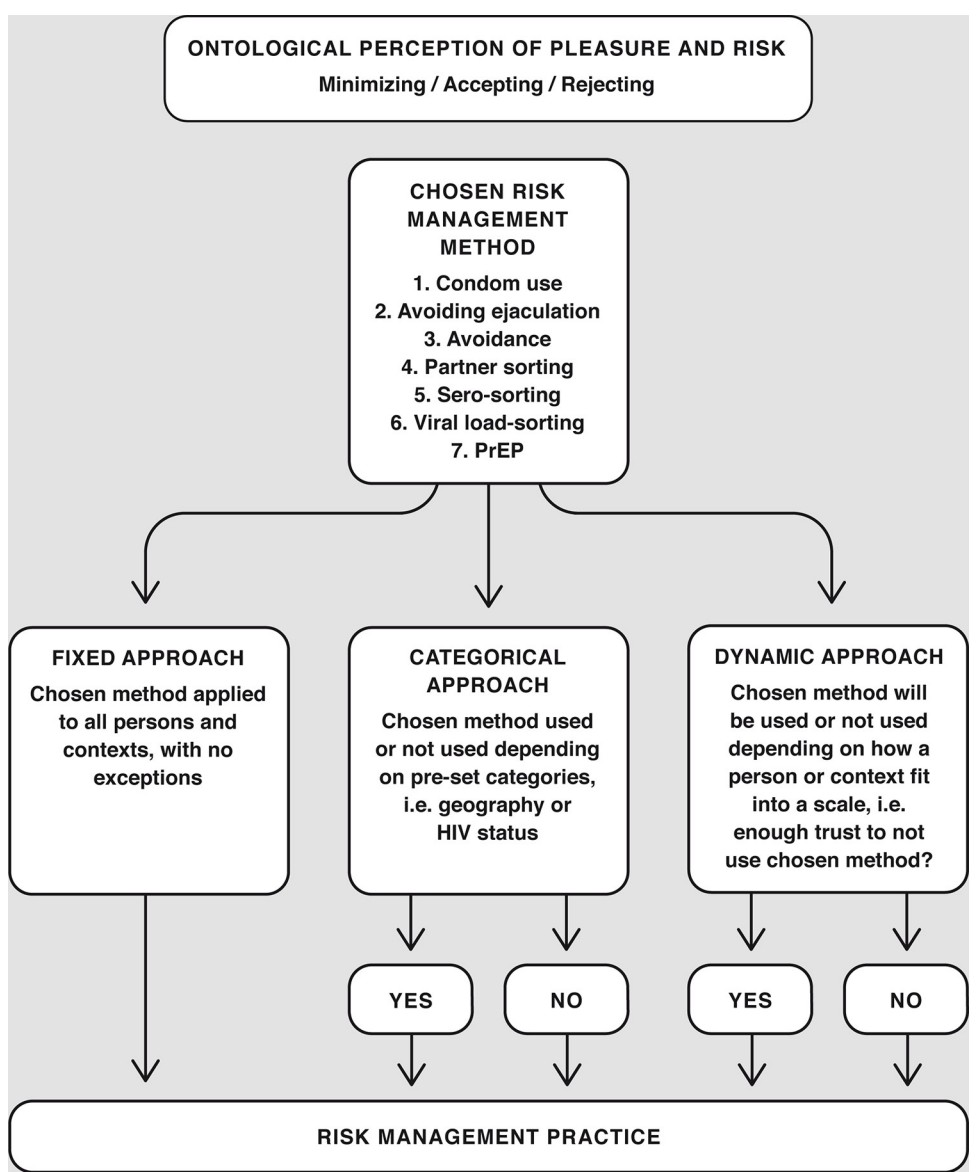

**Fig 1. Conceptualization model: The decision process for each risk management method chosen.**

decision-making process in a conceptual model below (Fig 1). Our interviews showed that accepting and rejecting risk could be more than deciding not to use a condom, as it was also an attitude of living without worries about seroconverting. While accepting risk could be a passive approach, the rejection of risk represents an active approach, a mind-set close to what Lupton and Tulloch describe about the civilized body and the will to explore full bodily pleasure [28]. Accepting and/or rejecting risk was manifested several times during the interviews and explained the decisions made by these men. The feeling of having one's body disciplined and controlled by public health messages and interventions generated resistance in the shape of choosing to not use a condom when having anal intercourse [37]. Previous studies have shown that doing so may not increase sexual pleasure, but rather the sense of being rebellious and chance-taking [36]. Rather than *fetishizing* risk, the men wanted to live without having to *relate* to risk.

The analyzed material provided the foundation for a conceptual model that improved our understanding of how safer sex decisions were made by including ontological perceptions as well as practical risk reduction methods mentioned by the informants (see model 1). The model suggests that *ontological perceptions of pleasure and risk* together form a risk management method that creates the starting point of the individual's decision-making process. This begins by defining the role and importance of sexual pleasure in relationship to ontological perceptions of risk, i.e. risk is something which must be rejected, accepted or minimized. Balancing pleasure and safety were key issues for the men interviewed and they asked themselves if a certain degree of pleasure was worth the risk it entailed [13]. The choice of risk management method/s and the decisions made on how to implement them depended on the ontological perception, creating the overall risk management practice.

The choice of approach provides keys to the next step, Risk management–chosen method, in which one of the risk management methods presented is inserted. If more than one method is applied the model replicates horizontally. Each method comes with one or more approaches, which suggests how the decisions are made. These three approaches are fixed, categorical and dynamic.

The *fixed approach* was an approach applied by MSM to all settings and partners, for example condom use on every occasion or daily regimen of PrEP. In contrast, those using a *categorical approach* had pre-set and defined categories to guide their decision making, for example event-based dosing of PrEP. One informant used pre-set spatial categories to decide when to use PrEP or not; he used PrEP in Berlin but not in Stockholm. Men who used a *dynamic approach* subjected themselves to inner negotiation before or during sex with new partners. The negotiation in the *dynamic approach* was not always based on the level of intimacy, trust or sexual arousal. It could also be related to different settings. The data indicated a fluidity between what was a private sex party at someone's home, a smaller fisting club for members only, the local sauna, techno clubs or commercial sex parties of all sizes. Differences were made between an open fisting night at a club where one has met almost everyone before and a private sex party found on a dating app. Thus, what appears to be a categorical decision based on space/context (darkroom, sex club or gallery opening) may actually be based on the level of trust with those present at a specific point in time.

We also found that some men described having parallel approaches to the same method depending on two or more parameters. For example, the first parameter was whether the partner's HIV status was known or not. If not known, a partner could be HIV negative or HIV positive with or without an undetectable viral load. This would determine whether condoms were used or not as a decision conditioned on pre-set categories. A second parameter could be the "degree of intimacy". This parameter guided the decision taken after an inner negotiation by asking 'is the level of intimacy high enough for me not to use a condom?'. This decision-making process was dynamic.

U = U and PrEP are defining a new era in HIV prevention, providing MSM with new possibilities to balance pleasure and safety. PrEP was known by the interviewees although it was not available from the Swedish healthcare system at the time of the interviews. By promoting U = U and/or PrEP as a way to reduce stigma, wellbeing may be noticeably reduced, although they will not protect these men from STI's. What was called 'HIV optimism '[73] is now integrated in prevention and in the mindset of the U = U message. In contrast, other MSM may prefer the 1980's mantra messages of "use condoms at intercourse and avoid semen in your mouth" [74].

## Conclusions

The proposed model includes three ontological perspectives on pleasure and risk, seven different risk management methods and three approaches on how to use them. It visualizes the complexity of decision making among a group of highly sexually active and knowledgeable MSM. A man who applies a single method and a fixed approach is easy to understand for himself and for public health professionals and prevention scientists. Illustrating the process for men with shifting ontological perceptions of pleasure and risk, multiple methods, parameters, categories and continuous inner negotiations requires additional work. Our model could be used for further discussions within the research and HIV prevention community in order to create deeper understanding of the challenges that MSM face in HIV/STI risk management and risk perception.

Our results also suggest that MSM with multiple sexual partners integrate the presence of risk into the very core of their sexuality, belonging to a community deeply affected by the HIV epidemic. When thinking about HIV before, during and after sex, there is a recurring struggle for balance between pleasure and safety. MSM have to deal with the fact that their sexuality and bodies are perceived as something intimately linked with HIV. Some men choose to reject this link, not only because condom use is less pleasurable, but also due to a profound rejection of the homosexual body as a potential vessel for an epidemic. The body could instead be "the garden of the soul" to quote the HIV/AIDS drama 'Angels in America' [18].

The implications for this can be seen on multiple levels, the most obvious one is the rejection of over-simplified safer sex messaging focusing solely on condom provision, testing and appealing to the rational, civilized body. Although community-based HIV prevention has traditionally aimed to include pleasure into HIV prevention, there are great limitations if the methods used today are the only methods which were available early on in the epidemic, without later additions to the preventive toolbox. Aligned with the wish among MSM to live without a sense of risk, PrEP also taps into other aspects of MSM's overall quality of life [13]. Some countries have been early adopters of PrEP policy and, to some extent, PrEP programming after continuous effort from different HIV prevention and anti-stigma organizations [14, 15]. Despite progress, global access to PrEP-programing is still behind, echoing late introduction of universal access to anti-retroviral treatment in the global south.

### Strengths, limitations and transferability

A strength of this study is that it was performed on-site in Berlin with one person conducting all of the interviews and leading the analysis. There is a known risk of social desirability bias when conducting interview-based research. Yet based on the informants' openness and willingness to describe their experiences with risk taking behaviors such as condomless anal sex, we did not notice indications of social desirability bias when conducting the interviews. However, we cannot exclude that there are people who refused to participate who have higher levels of risk taking or who may have different thoughts and experiences compared to the participants in this study.

The present study was limited to a highly mobile and sexually active, well-educated subgroup of MSM, a population disproportionately affected by HIV. Future studies should take a more intersectional perspective, include more data on broad experiences and risk factors and thereby allow for more in-depth analysis of risk behaviour. Also, future research should investigate the complexity of how ontological aspects relate to the decision-making process and additional methods of safety in other sexually active populations and sexual risks such as other STIs and unwanted pregnancy. The potential transferability of fixed, categorical and dynamic approaches may be studied with heterosexually identified men and women and pregnancy

risk, as well as among other sexually vulnerable populations such as transgender. Furthermore, we suggest that the conceptual model for decision making may also be relevant to other risk-taking behaviors. The COVID-19 pandemic has clearly shown that risk perception and risk management strategies vary between groups and individuals. The pandemic has also taught us that stigmatization and discrimination of groups of people can quickly take new forms. Health-related stigmatization of specific groups is not limited to MSM or sex. Public health strategies need to consider the effects of perceptions of power in populations considering themselves to be marginalised or "different" from the mainstream with regards to adopting "acceptable" ways of behaving.

## Supporting information

**S1 File. Interview guide SWE ENG.**
(DOCX)

## Acknowledgments

The authors wish to thank all the men who participated and shared their experiences. Also, we would like to extend our appreciation to Tobias Herder for on-going discussions on critical HIV prevention and to Olov Lindblad for reviewing the conceptualization and an early draft of the manuscript. We would also like to thank Mathilde Sengoelge for proofreading the manuscript.

## Author Contributions

**Conceptualization:** Nicklas Dennermalm, Kristina Ingemarsdotter Persson, Sarah Thomsen, Birger C. Forsberg.

**Data curation:** Nicklas Dennermalm.

**Formal analysis:** Nicklas Dennermalm.

**Funding acquisition:** Kristina Ingemarsdotter Persson, Birger C. Forsberg, Helle Mølsted Alvesson.

**Investigation:** Nicklas Dennermalm.

**Methodology:** Nicklas Dennermalm, Kristina Ingemarsdotter Persson, Sarah Thomsen, Helle Mølsted Alvesson.

**Supervision:** Kristina Ingemarsdotter Persson, Sarah Thomsen, Birger C. Forsberg, Helle Mølsted Alvesson.

**Writing – original draft:** Nicklas Dennermalm, Kristina Ingemarsdotter Persson.

**Writing – review & editing:** Nicklas Dennermalm, Kristina Ingemarsdotter Persson, Sarah Thomsen, Birger C. Forsberg, Helle Mølsted Alvesson.

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
