## [Decision Letter · Decision Letter 0]

15 Nov 2021

PGPH-D-21-00685

Conceptualizing safer sex in a new era: Risk perception and decision-making process among highly sexually active men who have sex with men

Dear Dr. Dennermalm,

Thank you for submitting your manuscript to PLOS Global Public Health. After careful consideration, we feel that it has merit but does not fully meet PLOS Global Public Health’s publication criteria as it currently stands. Therefore, we invite you to submit a revised version of the manuscript that addresses the points raised during the review process.

We look forward to receiving your revised manuscript.

Kind regards,

Maria del Mar Pastor Bravo, Ph.D.

Academic Editor

Journal Requirements:

1. Please include a copy of the interview guide used in the study, in both the original language and English, as Supporting Information, or include a citation if it has been published previously.

2. Please provide separate figure files in .tif or .eps format only, and remove any figures embedded in your manuscript file.  If you are using LaTeX, you do not need to remove embedded figures.

3. Tables should not be uploaded as individual files.  Please remove these files and include the tables in your manuscript file.

4. In the online submission form, you indicated that "No additional or primary data will be made available due to the sensitive nature of the data and in alignment with the decision of the Ethical Review Board in Stockholm. This is to ensure the anonymity of the participants."

5. Please amend your detailed Financial Disclosure statement. This is published with the article, therefore should be completed in full sentences and contain the exact wording you wish to be published.

i). State the initials, alongside each funding source, of each author to receive each grant.

ii). State what role the funders took in the study. If the funders had no role in your study, please state: “The funders had no role in study design, data collection and analysis, decision to publish, or preparation of the manuscript.”

Reviewers' comments:

Reviewer's Responses to Questions

**Comments to the Author**

1. Does this manuscript meet PLOS Global Public Health’s publication criteria? Is the manuscript technically sound, and do the data support the conclusions? The manuscript must describe methodologically and ethically rigorous research with conclusions that are appropriately drawn based on the data presented.

Reviewer #1: Yes

Reviewer #2: Yes

2. Has the statistical analysis been performed appropriately and rigorously?

Reviewer #1: N/A

Reviewer #2: N/A

3. Have the authors made all data underlying the findings in their manuscript fully available (please refer to the Data Availability Statement at the start of the manuscript PDF file)?

Reviewer #1: No

Reviewer #2: No

4. Is the manuscript presented in an intelligible fashion and written in standard English?

Reviewer #1: Yes

Reviewer #2: Yes

5. Review Comments to the Author

Reviewer #1: The is a well-written, thoughtful paper with important contributions towards helping improve our understanding of risk and pleasure among MSM. It could be strengthened by including intersectional analysis of how race, age, and other factors might affect perceptions of risk, and by making more concrete linkages to health promotion among MSM in the Discussion and Conclusion.

The Introduction is strong, with an excellent exploration of the literature and outlining of key issues. Can you try to bring in some global context at the beginning, to locate Sweden within the context of global health promotion on HIV/AIDS? E.g. why the choice of Sweden, how do efforts in Sweden compare to other settings, can you share data about prevalence in Sweden compared to other settings? Some statement on the state of existing qualitative research on MSM in Sweden might also be useful for the reader, to understand where the gaps lie.

Given the issue of pleasure arises in the results, it might be useful to foreground this a bit more in the introduction. Can you add additional literature on pleasure? Some suggestions may include Calabrese et al (2015) which focuses on race, Mabire et al (2019) which focuses on PrEP etc.

p. 7 – lines 150-152 – in the eligibility criteria, relationship status was not mentioned. Were all the men single? Were any married? This is quite important to clarify.

p. 7-8 – Berlin is mentioned a few times in the methods section. Can you explain the relevance of MSM’s presence in both settings? Is this a factor for a specific reason? The comparisons between Berlin and Sweden in the results would be clearer if this was explained earlier in the paper. Also, is there a reason it is always the city of Berlin compared with the whole country of Sweden?

In the methods section, some detail on ethics is needed apart from the ethics statement at the end of the paper. How was consent and confidentiality managed? Was data anonymised? Were referrals provided to participants who might have been distressed during the interview process, or who disclosed violence? How willing were participants to share details of other contacts and what ethical issues were associated with this sharing of information (referrals)?

Listing numbers in brackets after participant quotes is quite a clinical approach to presenting participant data which may be a bit dehumanising – can you find another way of presenting the information?

Are there other factors that might influence MSM’s perspectives, that could be drawn out in the results. Race, economic status, migration status, education level and age are all factors that intersect to complicate and nuance MSM’s experiences and might result in different perceptions of risk. Can you take a more intersectional lens to draw out these aspects when presenting the data?

p. 12 (line 259) – Did the participant describe this as being like the sword of Damocles or is this the authors interpretation? Can you clarify this statement?

The conclusion could go a little further in articulating the implications for health promotion. Can you build on this more in the conclusion? The Discussion section would also benefit from greater discussion of the implications for public health messages globally. What implications might this have for work with MSM in settings like the US or UK?

Reviewer #2: This is a well developed manuscript. The only comments are:

1. It would be useful to make use of the COREQ check list for qualitative data

2. Either in the methods or the discussion, the authors should address the issue of reflexivity and how it was addressed either in data collection, analysis or interpretation.

6. PLOS authors have the option to publish the peer review history of their article (what does this mean?). If published, this will include your full peer review and any attached files.

**Do you want your identity to be public for this peer review?** For information about this choice, including consent withdrawal, please see our Privacy Policy.

Reviewer #1: No

Reviewer #2: No

---

## [Editor Report · Decision Letter 1]

6 Apr 2022

Conceptualizing safer sex in a new era: Risk perception and decision-making process among highly sexually active men who have sex with men

PGPH-D-21-00685R1

Dear Mr Dennermalm,

We are pleased to inform you that your manuscript 'Conceptualizing safer sex in a new era: Risk perception and decision-making process among highly sexually active men who have sex with men' has been provisionally accepted for publication in PLOS Global Public Health.

Best regards,

Maria del Mar Pastor Bravo, Ph.D.

Academic Editor
